# VirtualPainting: Addressing Sparsity with Virtual Points and Distance-Aware Data Augmentation for 3D Object Detection

**DOI:** 10.3390/s25113367

**Published:** 2025-05-27

**Authors:** Sudip Dhakal, Deyuan Qu, Dominic Carrillo, Mohammad Dehghani Tezerjani, Qing Yang

**Affiliations:** 1Department of Computer Science and Engineering, University of North Texas, Denton, TX 76205, USA; deyuan.qu@unt.edu (D.Q.); dominic.carrillo@unt.edu (D.C.); mike.degany@unt.edu (M.D.T.); qing.yang@unt.edu (Q.Y.); 2Department of Software Engineering, Florida Gulf Coast University, Fort Myers, FL 33965, USA; 3Dendritic: A Human-Centered AI and Data Science Institute, Florida Gulf Coast University, Fort Myers, FL 33965, USA

**Keywords:** three-dimensional object detection, multimodal fusion, semantic segmentation, sparse object detection, occluded object detection

## Abstract

In recent times, there has been a notable surge in multimodal approaches that decorate raw LiDAR point clouds with camera-derived features to improve object detection performance. However, we found that these methods still grapple with the inherent sparsity of LiDAR point cloud data, primarily because fewer points are enriched with camera-derived features for sparsely distributed objects. We present an innovative approach that involves the generation of virtual LiDAR points using camera images and enhancing these virtual points with semantic labels obtained from image-based segmentation networks to tackle this issue and facilitate the detection of sparsely distributed objects, particularly those that are occluded or distant. Furthermore, we integrate a distance-aware data augmentation (DADA) technique to enhance the model’s capability to recognize these sparsely distributed objects by generating specialized training samples. Our approach offers a versatile solution that can be seamlessly integrated into various 3D frameworks and 2D semantic segmentation methods, resulting in significantly improved overall detection accuracy. Evaluation on the KITTI and nuScenes datasets demonstrates substantial enhancements in both 3D and bird’s eye view (BEV) detection benchmarks.

## 1. Introduction

3D object detection plays a pivotal role in enhancing scene understanding for safe autonomous driving. In recent years, a large number of 3D object detection techniques have been implemented [1,2,3,4,5,6,7]. These algorithms primarily leverage information from LiDAR and camera sensors to perceive their surroundings. LiDAR provides low-resolution shape and depth information [8], while cameras capture dense images rich in color and textures. While multimodal-based 3D object detection has made significant advancements recently, their performance still notably deteriorates when dealing with sparse point cloud data. Recently, painting-based methods like PointPainting [6] have gained popularity in an attempt to address this issue. These methods decorate the LiDAR points with camera features. However, a fundamental challenge still persists in the case of objects lacking corresponding point clouds, such as distant and occluded objects. Despite the presence of camera features for such objects, there is no associated point cloud to complement these features. As a result, although they are beneficial for improving overall 3D detection performance, they still contend with sparse point clouds, as illustrated in Figure 1.

In the context of autonomous driving, remote objects refer to those located beyond 30–50 m from the ego vehicle, often resulting in very few or no LiDAR points due to sensor limitations. Occluded objects are partially or fully blocked by other objects, such as parked vehicles or pedestrians behind large structures. These scenarios are common and pose critical challenges for object detection systems. Our goal is to enhance detection accuracy under these conditions by enriching sparse LiDAR inputs with semantically meaningful virtual points derived from camera images. In Figure 1, the yellow circle highlights a pedestrian that is occluded in the LiDAR view, resulting in the absence of 3D point cloud data at that location. The red bounding box shows a failure in detection caused by this sparsity, even though semantic information exists in the camera image. This illustrates a common failure case where conventional LiDAR-only or even painted-point approaches fall short due to the lack of geometric support.

One such practical implication of our model is risk anticipation and planning. In autonomous driving systems, while close-range detection is crucial for immediate response, early recognition of distant and occluded objects also plays a vital role in risk anticipation and planning, particularly at high speeds (e.g., highway scenarios) or in occluded urban intersections. For instance, recognizing a pedestrian or cyclist partially visible behind a parked vehicle from a distance allows autonomous systems to proactively adjust speed or trajectory, rather than react late. Thus, detecting sparse and remote objects is essential for both proactive safety and smoother navigation [7]. In response to the inherent sparsity of LiDAR data, several methods have been introduced to generate pseudo or virtual points. These methods bolster the sparse point clouds by introducing supplementary points around the existing LiDAR points. For example, Multimodal Virtual Point (MVP) [9] generates virtual points by completing the depth information of 2D instance points using data from the nearest 3D points. Similarly, Sparse Fuse Dense (SFD) [10] creates virtual points based on depth completion networks [11]. These virtual points play a crucial role in enhancing the geometric representation of distant objects, showcasing their potential to significantly enhance the performance of 3D detection methods. However, current implementations have yet to fully harness the advantages of integrating semantic results from semantic networks with virtual points. The incorporation of semantic information into the augmented point cloud, which includes both the original and virtual points, not only enriches the dataset but also increases the model’s overall robustness [6].

Most recent fusion-based methods primarily concentrate on different fusion stages: early fusion, which involves combining LiDAR and camera data at an early stage; deep fusion, where features from both camera and LiDAR are combined by feature fusion; and late fusion, which combines the output candidates or results from both LiDAR and camera detection frameworks at a later stage. However, there is minimal emphasis on the quality of the training data, a crucial aspect in any detection framework. This issue is particularly evident in many fusion-based methods that lack sufficient sparse training samples for sparsely distributed object such as occluded and distant objects. The absence of comprehensive training data makes the trained model fragile and, as a result, incapable of effectively detecting distant objects during the testing phase. Consequently, fusion-based methods also face the challenge of inadequate data augmentation. The inherent disparities between 2D image data and 3D LiDAR data make it difficult to adapt several data augmentation techniques that are effective for the latter. This limitation poses a significant barrier and is a primary factor leading to the generally lower performance of multimodal methods in comparison to their single-modal counterparts [12].

To address these issues, we propose a simple yet effective “VirtualPainting” framework. Our method addresses the problem of sparse LiDAR points by generating virtual points using depth completion network PENet [11]. To elaborate, we initiate the generation of supplementary virtual points and seamlessly merge them with the original points, resulting in an augmented LiDAR point cloud dataset. This augmented dataset subsequently undergoes a “painting” process utilizing features derived from cameras. The camera-derived features are in the form of semantic scores or per-pixel class scores. The augmented LiDAR point cloud is concatenated with per pixel class score to obtain a feature-rich point cloud. The result is twofold: it not only yields a denser point cloud in the form of augmented LiDAR point clouds but also enables a seamless combination of camera features and the point clouds. The virtual points, generated via the depth completion network, are now linked with camera features, resulting in a more comprehensive data representation. In certain scenarios where camera features were present but lacked corresponding LiDAR sensor points, these camera features remained unincorporated. Although high-resolution cameras provide rich semantic information, they lack precise depth data, which is critical for accurate 3D localization and object boundary estimation [7]. LiDAR complements this by offering geometrically accurate range information. Our proposed method leverages both modalities, particularly benefiting cases like those shown in Figure 1, where high-resolution semantic cues exist but lack corresponding depth data from LiDAR. Without LiDAR, these objects are semantically identifiable but not localizable in 3D space, which is essential for autonomous navigation.

Additionally, we address the challenges of insufficient training samples for distant objects and the absence of adequate data augmentation techniques by integrating a method called distance-aware data augmentation (DADA). In this approach, we intentionally generate sparse training samples from objects that are initially densely observed by applying a distant offset. Considering that real-world scenes frequently involve incomplete data due to occlusion, we also introduce randomness by selectively removing portions to simulate such occlusion. By integrating these training samples, our model becomes more resilient during the testing phase, especially in the context of detecting sparsely distributed objects, such as occluded or distant objects, which are frequently overlooked in many scenarios. Examples of objects that frequently suffer from sparse point coverage include pedestrians stepping out from behind vehicles, cyclists approaching from a distance, and vehicles at intersections over 50 m away. Our method enhances detection accuracy for these cases by generating virtual points where LiDAR fails and enriching them with semantic cues. In brief, our contributions are summarized as follows:An innovative approach that augments virtual points obtained through the joint application of LiDAR and camera data with semantic labels derived from image-based semantic segmentation.Integration of distance-aware data augmentation method for improving models’ ability to identify sparse and occluded objects by creating training samples.A generic method that can incorporate any variation of 3D frameworks and 2D semantic segmentation for improving the overall detection accuracy.Evaluation on the KITTI and nuScenes datasets shows major improvement in both 3D and BEV detection benchmarks, especially for distant objects.

## 2. Related Work

### 2.1. Single Modality

Existing LiDAR-based methods can be categorized into four main groups based on their data representation: point-based, grid-based, point-voxel-based, and range-based. PointNet [13] and PointNet++ [14] are early pioneering works that apply neural networks directly to point clouds. PointRCNN [15] introduced a novel approach, directly generating 3D proposals from raw point clouds. VoxelNet [1] introduced the concept of a VFE (voxel feature encoding) layer to learn unified feature representations for 3D voxels. Building upon VoxelNet, CenterPoint [16] devised an anchor-free method using a center-based framework based on CenterNet [17], achieving state-of-the-art performance. SECOND [2] harnessed sparse convolution [18] to alleviate the challenges of 3D convolution. PV-RCNN [19] bridged the gap between voxel-based and point-based methods to extract more discriminative features. Voxel-RCNN [3] emphasized that precise positioning of raw points might not be necessary, contributing to the efficiency of 3D object detection. PointPillars [12] innovatively extracted features from vertical columns (Pillars) using PointNet [13]. Despite these advancements, all these approaches share a common challenge—the inherent sparsity of LiDAR point cloud data, which impacts their overall efficiency.

### 2.2. Fusion-Based

The inherent sparsity of LiDAR point cloud data has sparked research interest in multimodal fusion-based methods. MV3D [4] and AVOD [20] create a multi-channel bird’s eye view (BEV) image by projecting the raw point cloud into BEV. AVOD [20] takes both LiDAR point clouds and RGB images as input to generate features shared by the Region Proposal Network (RPN) and the refined network. MMF [21] benefits from multi-task learning and multi-sensor fusion, while 3D-CVF [5] fuses features from multi-view images, and Sniffer Faster RCNN [22] combines and refines the 2D and 3D proposal together at the final stage of detection. CLOCs [23] and Sinffer Faster R-CNN++ [24] take it one step further and refine the confidences of 3D candidates using 2D candidates in a learnable manner. These methods face limitations in utilizing image information due to the sparse correspondences between images and point clouds. Additionally, fusion-based methods encounter another challenge—a lack of sufficient data augmentation. In this paper, we address both issues by capitalizing on the virtual points and using data augmentation techniques.

### 2.3. Point Decoration Fusion

Recent developments in fusion-based methods have paved the way for innovative approaches like point decoration fusion-based methods, which enhance LiDAR points with camera features. PointPainting [6], for instance, suggests augmenting each LiDAR point with the semantic scores derived from camera images. PointAugmenting [7] acknowledges the limitations of semantic scores and proposes enhancing LiDAR points with deep features extracted from a 2D object detectors. FusionPainting [25] takes a step further by harnessing both 2D and 3D semantic networks to extract additional features from both camera images and the LiDAR point cloud. Nevertheless, it is important to note that these methods also grapple with the sparse nature of LiDAR point cloud data, as illustrated in Figure 1. Even though PointAugmenting addresses this issue with data augmentation techniques, the inadequacy of sparse training samples for sparse objects leads to their failure in detection.

## 3. VirtualPainting

We provide an overview of the proposed “VirtualPainting” framework in Figure 2. It addresses the sparsity of LiDAR point clouds and the lack of training samples for sparse objects by introducing multiple enhancement modules. These include semantic segmentation, depth completion, virtual point painting, and distance-aware data augmentation, ultimately feeding into a 3D object detection network. For details on each stage, please refer to Figure 2. Each component plays a specific role in improving the quality and density of point cloud data. Together, they contribute to more reliable detection of distant and occluded objects in challenging driving scenarios.

### 3.1. Image-Based Semantic Network

Two-dimensional images captured by cameras are rich in texture, shape, and color information. This richness offers valuable complementary information for point clouds, ultimately enhancing three-dimensional detection. To leverage this synergy, we employ a semantic segmentation network to generate pixel-wise semantic labels. We employ the BiSeNetV2 [26] segmentation model for this purpose. This network takes multi-view images as its input and delivers pixel-wise classification labels for both foreground instances and the background. It is worth noting that our architecture is flexible, allowing for the incorporation of various semantic segmentation networks to generate semantic labels. For semantic segmentation in our KITTI dataset experiments, we use the BiSeNetv2 [26] network. This network underwent an initial pretraining phase on the CityScapes [27] dataset and was subsequently fine-tuned on the KITTI dataset using PyTorch 2.0 [28]. For the simplicity of our implementation, we chose to ignore classes cyclist and bike because there is a difference in the class defination of a cyclist between KITTI semantic segmentation and object detection tasks. In object detection, a cyclist is defined as a combination of the rider and the bike, whereas in semantic segmentation, a cyclist is defined as solely the rider, with the bike being a separate class. Similarly for nuScenes, we developed a custom network using the nuImages dataset, which comprises 100,000 images containing 2D bounding boxes and segmentation labels for all nuScenes classes. The segmentation network utilized a ResNet backbone to extract features at various strides, ranging from 8 to 64, and incorporated an FCN segmentation head for predicting nuScenes segmentation scores.

### 3.2. PENet for Virtual Point Generation

The geometry of nearby objects in LiDAR scans is often relatively complete, whereas for distant objects, it is quite the opposite. Additionally, there is a challenge of insufficient data augmentation due to the inherent disparities between 2D image data and 3D LiDAR data. Several data augmentation techniques that perform well with 3D LiDAR data are difficult to apply in multimodal approaches. This obstacle significantly contributes to the usual under performance of multimodal methods when compared to their single-modal counterparts. To address these issues, we employ PENet to transform 2D images into 3D virtual point clouds. This transformation unifies the representations of images and raw point clouds, allowing us to handle images much like raw point cloud data. We align the virtual points generated from the depth completion network with the original points to create a augmented point cloud data. This approach collectively enhances the geometric information of sparse objects while also establishing an environment for a unified representation of both images and point clouds. Much like SFD, our approach relies on the virtual points produced by PENet. The PENet architecture is initially trained exclusively on the KITTI dataset, which encompasses both color images and aligned sparse depth maps. These depth maps are created by projecting 3D LiDAR points onto their corresponding image frames. The dataset comprises 86,000 frames designated for training, in addition to 7000 frames allocated for validation, and a further 1000 frames designated for testing purposes. Our PENEt is trained on the training set.The images are standardized at a resolution of 1216 × 352. Typically, a sparse depth map contains approximately 5 valid pixels, while ground truth dense depth maps have an approximate 16 coverage of valid pixels [11].

### 3.3. Painting Virtual Points

The current implementation of LiDAR point cloud painting methods has not harnessed the advantages of associating semantic results from the semantic network with virtual points. Incorporating semantic information into the augmented point cloud, which includes both the original and virtual points, not only enriches the dataset but also enhances the model’s robustness. Let us refer to the original points generated from LiDAR scan as “raw point cloud”, denoted by *R*, and the point clouds generated from depth completion network as “virtual points”, denoted by *V*. Starting with a set of raw clouds *R*, we have the capability to transform it into a sparse depth map *S* using a known projection $T_{\mathrm{LiDAR}\to \mathrm{image}}$. We also have an associated image denoted as *I* corresponding to *R*. By providing both *I* and *S* as inputs to a depth completion network, we obtain a densely populated depth map labeled as *D*. Utilizing a known projection $T_{\mathrm{image}\to \mathrm{LiDAR}}$, we can then generate a set of virtual points denoted as *V*.

Our VirtualPainting algorithm comprises three primary stages. In the initial stage, utilizing the virtual points acquired from the depth completion network, we align these virtual points with the original raw LiDAR points, effectively generating an augmented LiDAR point cloud denoted as *A* with N points. In the second stage, as previously mentioned, the segmentation network produces C-class scores. In the case of KITTI [29], C equals 4 (representing car, pedestrian, cyclist, and background), while for nuScenes [30], C is 11 (comprising 10 detection classes along with background). In the final stage, the augmented LiDAR points undergo projection onto the image, and the segmentation scores corresponding to the relevant pixel coordinates (h, w) are appended to the augmented LiDAR point, resulting in the creation of a painted LiDAR point. This transformation process involves a homogeneous transformation followed by projection into the image as given in Algorithm 1. Algorithm 1 outlines the process of generating painted augmented LiDAR points. Here is a step-by-step explanation of each input and operation:$L\in R^{N,D}$ denotes the original raw LiDAR point cloud, where *N* is the number of points and D≥3 indicates spatial dimensions (e.g., x,y,z).$V\in R^{N,D}$ represents the virtual points generated using a depth completion network. These are spatially aligned with the original LiDAR points.The augmented point cloud *A* is obtained by combining the raw and virtual points, i.e., A=L+V, where $A\in R^{N,D}$.$S\in R^{W,H,C}$ is the semantic segmentation score map, where *W* and *H* are the width and height of the image, and *C* is the number of semantic classes.$T\in R^{4\times 4}$ is the homogeneous transformation matrix used to project 3D points into the camera coordinate frame.$M\in R^{3\times 4}$ is the camera projection matrix used to map 3D coordinates into 2D image coordinates.

The output is $P\in R^{N,D+C}$, the painted augmented point cloud containing both spatial and semantic features.

The algorithm iterates over each point $\tilde{l}\in A$:1The 3D point $\tilde{l}$ is projected onto the image plane to obtain 2D coordinates ${\tilde{l}}_{\mathrm{image}}$ using the ‘PROJECT’ function, which applies the transformation *T* followed by projection through *M*.2The semantic class score vector $\tilde{s}\in R^{C}$ is retrieved from *S* at the pixel coordinates $\left({\tilde{l}}_{\mathrm{image}}\left[0\right],{\tilde{l}}_{\mathrm{image}}\left[1\right]\right)$.3The final painted point $\tilde{p}\in R^{D+C}$ is generated by concatenating the 3D point $\tilde{l}$ with its corresponding class score vector $\tilde{s}$.

This painted representation enriches the spatial LiDAR data with semantic context from the image, enabling better detection of sparse or distant objects.
**Algorithm 1:** VirtualPainting (L, V, A, S, T, M).
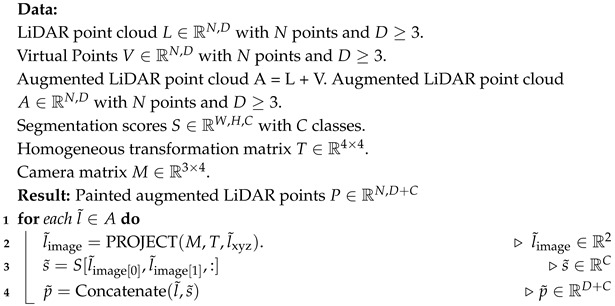


### 3.4. Distance-Aware Data Augmentation

As mentioned earlier, the absence of comprehensive geometric information for sparse objects can significantly hamper detection performance. To overcome this challenge, we aim to enhance our understanding of the geometry of sparsely observed distant objects by generating training samples derived from densely observed nearby objects. While several established methods exist to address this challenge, such as random sampling or farthest point sampling, it is important to note that these techniques often result in an uneven distribution pattern within the LiDAR-scanned point cloud. In this context, we adopt a sampling strategy [31] that takes into account both LiDAR scanning mechanics and scene occlusion. Within the context of a nearby ground truth box with position $C_{g}$ and its associated inside points ${\left\{P_{gi}\right\}}_{i}$, we introduce a random distance offset Δα as follows: $C_{g}:=C_{g}+\Delta \alpha ,P_{gi}:=P_{gi}+\Delta \alpha$. Subsequently, we proceed to convert the points ${\left\{P_{gi}\right\}}_{i}$ into a spherical coordinate system and voxelize them into spherical voxels, aligning with the LiDAR’s angular resolution. Within each voxel, we compute the distances between the points. If the points are found to be in very close proximity, with their distance being almost negligible, falling below a predefined threshold λ, we choose to calculate the average of these points. This results in a set of sampled points that closely mimic the distribution pattern of real scanned points, as depicted in Figure 2. During the training process, similar to the GT-AUG (Shi, Wang, and Li, 2019) [15] approach, we incorporate these sampled points and bounding box information into the training samples to facilitate data augmentation. This augmentation technique has the potential to address the shortage of training samples for distant objects. Additionally, we randomly remove portions of dense LiDAR points to simulate occlusion, aiming to potentially resolve the scarcity of occluded samples during training. In the last phase of our architecture, the 3D detector receives the input in the form of the painted version of the augmented point cloud. As there are no alterations to the backbones or other architectural components, providing the painted point cloud as input to any 3D detector such as PointRCNN, VoxelNet, PVRCNN, PointPillars, and so on, is very straightforward to obtain the final detection results. For our final experimental evaluation, we primarily used PV-RCNN due to its strong baseline performance on KITTI and nuScenes. However, we also tested on PointPillars and PointRCNN to show the generalizability of our approach.

### 3.5. LiDAR Network Details

We use the OPENPCDet [32] tool for the KITTI dataset, incorporating 3D detectors such as PVRCNN, PointPillars, PointRCNN, and VoxelNEt, with only minimal adjustments to the point cloud dimension. In order to accommodate segmentation scores from a semantic network, we expand the dimension by adding the total number of classes used in the segmentation network. Since our architecture remains unchanged beyond this point, it remains generic and can be readily applied to any 3D detector without requiring complex configuration modifications. Similarly, for the nuScenes dataset, we utilize an enhanced version of PointPillars, as detailed in [6]. These enhancements involve alterations to the pillar resolution, network architecture, and data augmentation techniques. Firstly, we reduce the pillar resolution from 0.25 m to 0.2 m to improve the localization of small objects. Secondly, we revise the network architecture to incorporate additional layers earlier in the network. Lastly, we adjust the global yaw augmentation from π to $\frac{\pi }{6}$ [6].

## 4. Experimental Setup and Results

We evaluate the effectiveness of our proposed VirtualPainting framework using two standard large-scale 3D object detection datasets: KITTI and nuScenes. The KITTI dataset comprises approximately 7500 training and 7500 testing frames, featuring objects such as cars, pedestrians, and cyclists. Detection difficulty is categorized into easy, moderate, and hard, based on object occlusion, truncation, and distance. The dataset exhibits a long-tail distribution with many pedestrians, cars and cyclists located beyond 30 m, contributing to severe point sparsity in LiDAR scans. Each sample includes front-view RGB images and Velodyne 3D point clouds, with accurate calibration between modalities. The nuScenes dataset includes 1000 scenes (20 s each) captured at 2 Hz across urban environments, and comprises over 1.4M annotated 3D bounding boxes across 10 classes including car, truck, pedestrian, bicycle, and motorcycle. It contains data from six cameras, one LiDAR, and five radars, offering full 360-degree coverage (see Figure 3). Its densely annotated frames, complex city scenes, and large class imbalance introduce challenging cases of sparsity and occlusion, making it ideal for evaluating robust multimodal methods like ours. While Table 1, Table 2 and Table 3 present our method’s quantitative improvements, we also include qualitative visualizations in Figure 4. These examples highlight successful detections of occluded or distant objects by our VirtualPainting-based models that were missed by baseline approaches, demonstrating our method’s robustness in sparse detection scenarios.

### 4.1. KITTI Results

We initially assess our model’s performance using the KITTI dataset and contrast it with the current state-of-the-art methods. In Table 1, you can observe the outcomes for the KITTI test BEV detection benchmark. Notably, there is a substantial enhancement in mean average precision (mAP) compared to both single and multimodality baseline methods like AVOD, MV3D, PointRCNN, PointPillars, and PVRCNN. This improvement is particularly prominent in the pedestrian and cyclist categories, which are more challenging to detect, especially when it comes to sparse objects. Notably, the moderate and hard classes within the pedestrian and cyclist categories exhibit more significant improvements when compared to the easy class, as depicted in Table 1. We can clearly observe that for PointRCNN, our approach exhibits a notable enhancement of +4.09, +3.79 and +4.52, +2.89, specifically for the moderate and hard difficulty levels within the pedestrian and cyclist classes. This trend is consistent across various other models as well. Similarly, Table 2 provides a comparison of our methods with point-painting based approaches. Although there is not a remarkable increase in mean Average Precision, there is still some notable enhancement.

### 4.2. nuScenes Results

Additionally, we assess the performance of our model using the nuScenes dataset, as displayed in Table 3. Our approach surpasses the single-modality PointPillar-based model in all categories. Furthermore, the enhanced variant called PaintedPointPillars, which is based on the point-painting method for PointPillars, exhibits improvements in terms of nuScenes Detection Score (NDS) and Average Precision (AP) across all ten classes in the nuScenes dataset. Similarly, as seen in our previous results, this improvement is most noticeable in classes such as pedestrian, bicycle, and motorcycle, where the likelihood of remaining undetected in occluded and distant sparse regions is higher.

## 5. Ablation Studies

### 5.1. VirtualPainting Is a Generic and Flexible Method

As depicted in Table 1 and Table 2, we assess the genericity of our approach by integrating it with established 3D detection frameworks. We carry out three sets of comparisons, each involving a single-modal method and its multimodal counterpart. The three LiDAR-only models under consideration are PointPillars, PointRCNN, and PVRCNN. As illustrated in Table 2, VirtualPainting consistently demonstrates enhancements across all single-modal detection baselines. These findings suggest that VirtualPainting possesses a general applicability and could potentially be extended to other 3D object detection frameworks. Likewise, Table 4 and Table 5 demonstrate that several elements, including virtual points and semantic segmentation, can be seamlessly incorporated into the architecture without requiring intricate modifications, thus highlighting the flexibility of our approach. Similary, we also evaluated the inference speed of our method on an NVIDIA RTX 3090 GPU. While the inference speed is higher compared to the original PointPainting method, it still remains faster than other single-modality based methods, as indicated in Table 5.

### 5.2. Where Does the Improvement Come from?

To gain a comprehensive understanding of how VirtualPainting leverages camera cues and augmentation techniques applied to LiDAR points to enhance 3D object detection models, we offer a thorough analysis encompassing both qualitative and quantitative aspects. Initially, we categorize objects into three groups based on their proximity to the ego-car: those within 30 m, those falling within the 30 to 50-m range, and those located beyond 50 m. Figure 5 and Figure 6 illustrate the relative improvements achieved through multimodal fusion within each of these distance groups. In essence, VirtualPainting consistently enhances accuracy across all distance ranges. Notably, it delivers significantly higher accuracy gains for long-range objects compared to short-range ones. This phenomenon may be attributed to the fact that long-range objects often exhibit sparse LiDAR point coverage, and the combination of augmentation techniques applied to these sparse points, rendering them denser, along with the inclusion of high-resolution camera semantic labels, effectively bridges the information gap. We use 15 m intervals (0–15, 15–30, etc.) to match typical LiDAR resolution drop-off ranges. The numbers of cyclist/pedestrian samples per bin in KITTI are approximately 0–15 m: 860; 15–30 m: 720; 30–45 m: 510; 45–60 m: 260. These distributions align with expected real-world object distributions across distances. Likewise, as illustrated in Table 4 and Table 6, the incorporation of semantic segmentation significantly elevates precision compared to other elements, including the virtual point cloud and DADA. Although there is noticeable improvement when both virtual points and DADA are integrated, the primary contribution stems from the attachment of semantic labels to the augmented point cloud, which contains both virtual points and the original LiDAR points for both KITTI and nuScenes datasets.

## 6. Conclusions

In this paper, we propose a generic 3D object detector that combines both LiDAR point cloud and camera image to improve the detection accuracy, especially for sparsely distributed objects. We address the sparsity and inadequate data augmentation problems of LiDAR point cloud through the combined application of camera and LiDAR data. Using a depth completion network, we generate virtual point clouds to make the LiDAR point clouds dense and adopt the point decoration mechanism to decorate the augmented LiDAR point clouds with image semantics, thus improving the detection for those objects that generally go undetected due to sparse LiDAR points. Moreover, we design a distance-aware data augmentation technique to make the model robust for occluded and distant objects. Experimental results demonstrate that our approach can significantly improve detection accuracy, particularly for these specific objects.

## Figures and Tables

**Figure 1 sensors-25-03367-f001:**
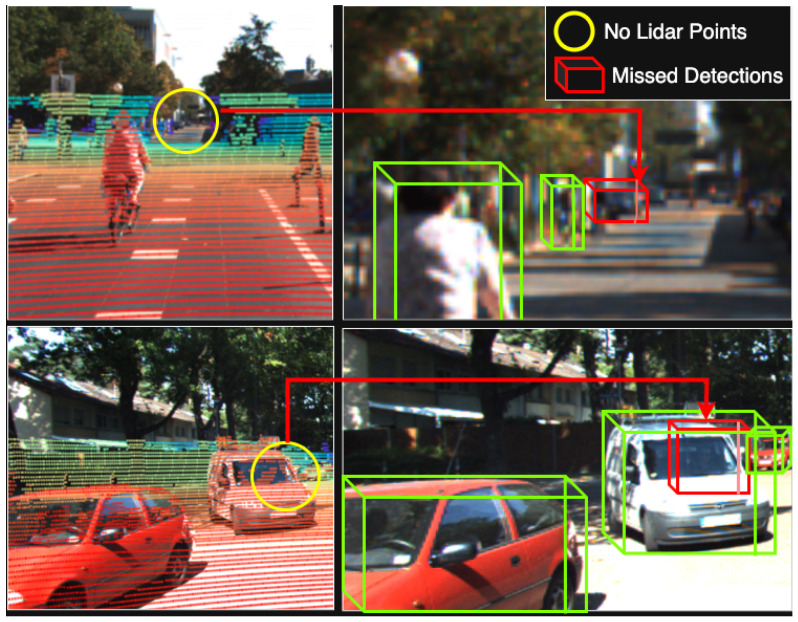
Drawback of painting-based methods [6]. The yellow circle serves as an indicator of the absence of point clouds in image projection, while the red bounding box signifies the consequent detection failure resulting from the sparse point cloud. Despite the presence of certain semantic cues for the object within the yellow circle, the absence of LiDAR points prevents the integration of these semantic cues with LiDAR data.

**Figure 2 sensors-25-03367-f002:**
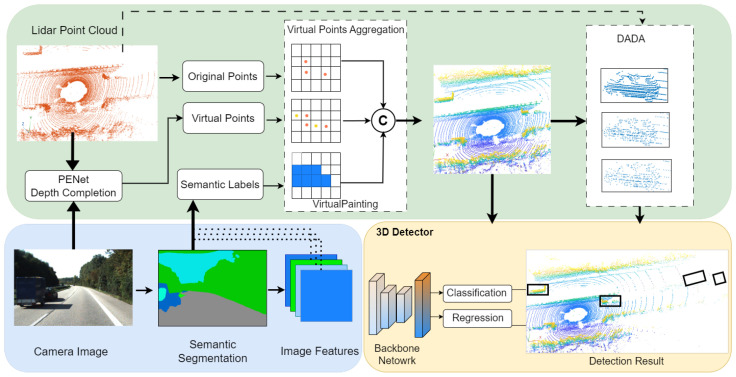
Overview of our VirtualPainting architecture, which comprises five distinct stages: firstly, a 2D semantic segmentation module responsible for computing pixel-wise segmentation scores; secondly, an image-based depth completion network named “PENet” generates the LiDAR virtual point cloud; thirdly, the VirtualPainting process involves painting virtual and original LiDAR points with semantic segmentation scores; and fourthly, the distance-aware data augmentation (DADA) component employs a distance-aware sampling strategy, creating sparse training samples primarily from nearby dense objects and, finally, a 3D detector for obtaining final detection result.

**Figure 3 sensors-25-03367-f003:**
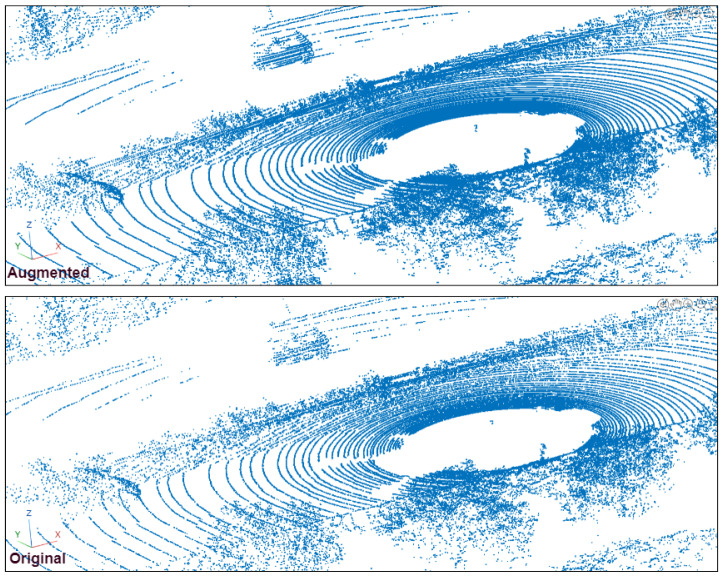
Illustration of augmented LiDAR point cloud and original LiDAR point cloud. The augmented LiDAR point cloud being slightly more dense in comparison to the original is the result of combining the original raw LiDAR points with the virtual points generated by PENet.

**Figure 4 sensors-25-03367-f004:**
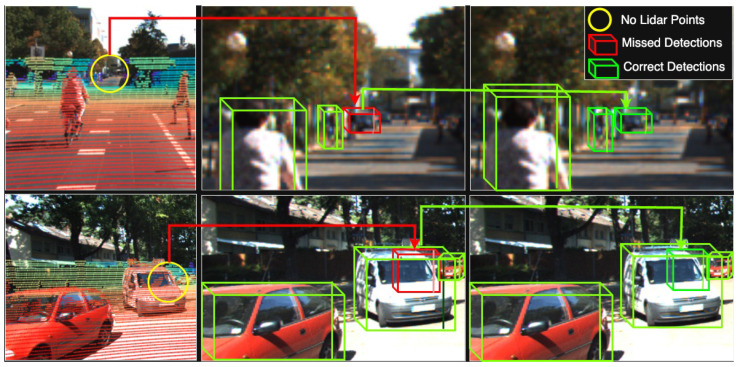
Qualitative comparison of object detection results. The baseline method fails to detect the distant object highlighted in the scene, while our proposed VirtualPainting approach successfully identifies it, demonstrating improved robustness for sparse and occluded objects.

**Figure 5 sensors-25-03367-f005:**
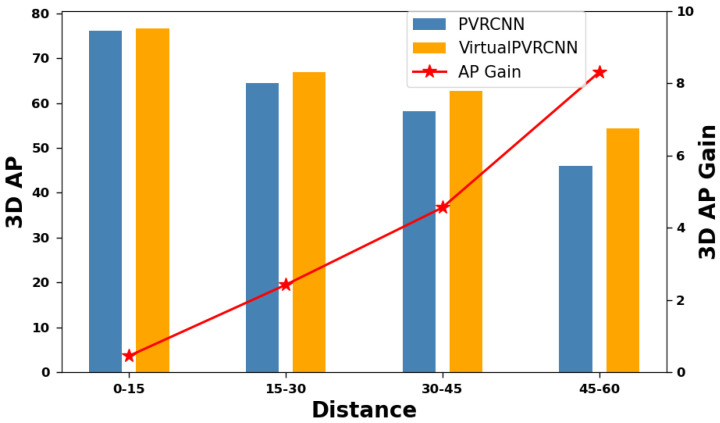
Performance improvement along different detection distance (KITTI test set) and 3D AP for cyclist class.

**Figure 6 sensors-25-03367-f006:**
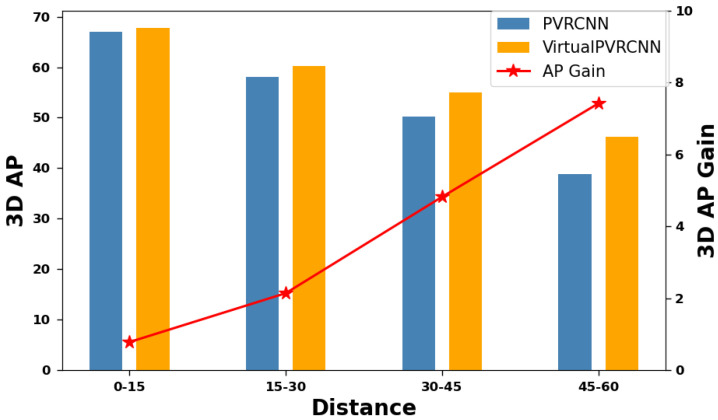
Performance improvement along different detection distance (KITTI test set) and 3D AP for pedestrian class.

**Table 1 sensors-25-03367-t001:** Results on the KITTI test BEV detection benchmark. L&I denotes LiDAR and Image input, i.e., multimodal models. “Improvement” refers to the increase in Average Precision (AP) of our VirtualPainting-based multimodal models relative to their corresponding LiDAR-only baselines. Performance gains are most evident for pedestrians and cyclists, particularly at longer distances where sparse LiDAR points often lead to missed detections. ↑ denotes additional improvement over baseline methods.

Method	Modality	mAP	Car			Pedestrian			Cyclist		
		**Mod.**	**Easy**	**Mod.**	**Hard**	**Easy**	**Mod.**	**Hard**	**Easy**	**Mod.**	**Hard**
AVOD	L&I	64.07	90.99	84.82	79.62	58.49	50.32	46.98	69.39	57.12	51.09
MV3D	L&I	N/A	86.62	78.93	69.80	N/A	N/A	N/A	N/A	N/A	N/A
PointRCNN	L	66.92	92.13	87.39	82.72	54.77	46.13	42.84	82.56	67.24	60.28
VirtualPointRCNN	L&I	70.56	92.67	88.35	83.67	58.99	50.22	46.63	84.07	71.76	63.17
Improvement ↑		+3.64	+0.54	+0.96	+0.95	+4.22	+4.09	+3.79	+1.51	+4.52	+2.89
PointPillars	L	65.98	90.07	86.56	82.81	57.60	48.64	45.78	79.90	62.73	55.58
VirtualPointPillars	L&I	69.07	90.39	86.98	83.59	58.86	50.03	46.71	80.15	65.89	59.44
Improvement ↑		+3.09	+0.32	+0.42	+0.78	+1.26	+1.39	+0.93	+0.25	+3.16	+3.86
PVRCNN	L	68.54	91.91	88.13	85.40	52.41	44.83	42.57	79.60	64.46	57.94
VirtualPVRCNN	L&I	70.05	92.33	88.64	86.08	53.57	48.97	46.76	80.26	68.39	61.91
Improvement ↑		+1.51	+0.42	+0.51	+0.68	+1.16	+4.14	+4.19	+0.66	+3.93	+3.97

**Table 2 sensors-25-03367-t002:** Comparision of PointPainting-based models with our VirtualPainting-based models.

**Model**	**Modality**	mAP			**Total** **mAP**	**Improvement**
		**Car**	**Pedestrian**	**Cyclist**		
PaintedPointRCNN	L&I	87.97	51.97	72.80	70.91	
VirtualPointRCNN	L&I	88.23	52.61	72.66	71.16	+0.25
PaintedPointPillars	L&I	87.18	51.44	68.18	69.02	
VirtualPointPillars	L&I	87.32	51.87	68.49	69.23	+0.21
PaintedPVRCNN	L&I	88.91	48.78	69.97	69.22	
VirtualPVRCNN	L&I	89.35	49.43	70.08	69.98	+0.76

**Table 3 sensors-25-03367-t003:** Comparisons of performance on the nuScenes test set, with reported metrics including NDS, mAP, and class-specific AP.

Method	Modality	NDS	mAP	Car	Truck	Bus	Trailer	Cons	Ped	Moto	Bicycle	TC	Barrier
3DCVF	L & C	62.3	52.7	83.0	45.0	48.8	49.6	15.9	74.2	51.2	30.4	62.9	65.9
PointPillars	L	55.0	40.1	76.0	31.0	32.1	36.6	11.3	64.0	34.2	14.0	45.6	56.4
PaintedPointPillars	L & C	58.1	46.4	77.9	35.8	36.2	37.3	15.8	73.3	41.5	24.1	62.4	60.2
VirtualPointPillars	L & C	60.21	48.59	79.20	36.11	39.47	40.53	16.27	76.25	47.45	28.79	65.71	63.77
Improvement ↑	–	+2.11	+2.19	+1.30	+0.29	+3.07	+3.23	+0.47	+2.95	+5.95	+4.69	+3.31	+3.57

**Table 4 sensors-25-03367-t004:** Analysis of the impacts of different components on the KITTI test set, with results assessed using the AP (Average Precision) metric calculated across 40 recall positions specifically for pedestrian and cyclist class.

Model	VPC	Sem.Seg	DA-Aug.	3D AP	
				Pedestrian	Cyclist
PointRCNN				47.85	68.93
VirtualPointRCNN	√		√	48.43	69.39
	√	√		51.75	72.04
	√	√	√	52.61	72.66
PointPillar				50.13	66.16
VirtualPointPillar	√		√	50.59	66.62
	√	√		51.45	68.06
	√	√	√	51.87	68.49
PVRCNN				46.41	67.26
VirtualPVRRCNN	√		√	46.83	67.73
	√	√		48.94	69.48
	√	√	√	49.43	70.08

**Table 5 sensors-25-03367-t005:** Ablation studies for different components on nuScenes dataset. Similar to KITTI, the semantic network plays a pivotal role in enhancing performance in the nuScenes dataset.

	VPC	Sem.Seg	DADA	mAP	NDS
(a)				40.12	55.02
(b)	√		√	41.03 + 0.93	55.75 + 0.75
(c)		√	√	47.05 + 6.75	58.73 + 2.98
(d)	√	√		47.78 + 0.73	59.59 + 0.86
(e)	√	√	√	48.59 + 0.81	60.21 + 0.62

**Table 6 sensors-25-03367-t006:** Inference speed across various multi- and single-modality frameworks.

VirtualPainting	PointPainting	F-PointNet	EPNet	3D-CVF
5.37FPS	2.38FPS	5.52FPS	8.15FPS	13.83FPS

## Data Availability

The data presented in this study are openly available in [Arxiv] at [https://doi.org/10.48550/arXiv.2312.16141].

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
