# Peer review of "VirtualPainting: Addressing Sparsity with Virtual Points and Distance-Aware Data Augmentation for 3D Object Detection"

_sensors, 2025, doi:10.3390/s25113367_

Round 1

Reviewer 1 Report

Comments and Suggestions for Authors

Please see the comment attachment.

Author Response

We sincerely thank the reviewer for their detailed and thoughtful feedback. We have addressed each concern carefully to improve the clarity, methodological rigor, and structure of the manuscript. Our detailed responses and revisions are provided below.

Comment 1:
Introduction Section. What are the practical implications of recognizing sparse targets at long distances? In autonomous driving, the greater concern is the accuracy of target recognition at close range. As the vehicle approaches, the long-range targets will also approach the vehicle and the features will be significant. Please describe the practical scenarios in which the problem is relevant and make them clear.

Response 1:
We agree that this clarification strengthens the motivation. In the revised Introduction (Page 2, Line 45), we added:

“While close-range detection is crucial for immediate response, early recognition of distant and occluded objects plays a vital role in risk anticipation and planning, particularly at high speeds (e.g., highway scenarios) or in occluded urban intersections. For instance, recognizing a pedestrian or cyclist partially visible behind a parked vehicle from a distance allows autonomous systems to proactively adjust speed or trajectory, rather than react late. Thus, detecting sparse and remote objects is essential for both proactive safety and smoother navigation.”

Comment 2:
Method - 3.3. Painting Virtual Points. The methodology is a very important section of the paper, but the authors do not elaborate enough on important derivation details (LN 179–188). Figure 3 is the result, not the method flow. Please put modeling results with quantitative statements (not qualitative) in the Results section.

Response 2:
Thank you for the feedback. We added a formal explanation of Algorithm 1 and clarified each step with mathematical detail in Section 3.3, Line 224. Figure 3 has been moved to the Results section as recommended. The added explanation is:

“Algorithm 1 outlines the process of generating painted augmented LiDAR points. We provide a step-by-step breakdown of each input variable and operation… [Full explanation inserted, including projections, concatenation steps, and output formation as detailed in the revised manuscript].”

This can be found in Section 3.3, Lines 226–248.

Comment 3:
Section 4. 3D Detector. Please merge this single and short section into Section 3. In addition, which one did 3D Detector ultimately pick?

Response 3:
We merged Section 4 into Section 3 as a concluding paragraph, Line 273. The revised text reads:

“In the last phase of our architecture, the 3D detector receives the input in the form of the painted version of the augmented point cloud… For our final experimental evaluation, we primarily used PV-RCNN due to its strong baseline performance on KITTI and nuScenes. However, we also tested on PointPillars and PointRCNN to show the generalizability of our approach.”

Comment 4:
Section 5: KITTI and nuScenes datasets should be briefly described (object class, sample size, distance distribution). Include performance pictures. “There is a greater likelihood of remaining undetected…” needs visual support.

Response 4:
We have added the following paragraph to the beginning of Section 5:

“We evaluate the effectiveness of our proposed VirtualPainting framework using two standard large-scale 3D object detection datasets: KITTI and nuScenes… While Table 1, Table 2 and Table 3 present our method’s quantitative improvements, we also include qualitative visualizations in Fig. 4. These examples highlight successful detections of occluded or distant objects by our VirtualPainting-based models that were missed by baseline approaches.”

Figures were moved and relabeled accordingly.

Comment 5:
Section 5.1–5.3 should be moved to Section 3 to consolidate methodology.

Response 5:
We sincerely thank the reviewer for their thoughtful suggestion regarding the organization of Sections 5.1–5.3. We understand the importance of clearly consolidating methodological components in a single section; however, we respectfully believe that the current structure serves a specific purpose and maintains a logical narrative flow in the paper.
Section 3 is dedicated to presenting the overall VirtualPainting architecture in a structured and comprehensive manner. It focuses on the conceptual design and technical flow of the proposed approach—explaining how various components such as semantic segmentation, depth completion, virtual point painting, and data augmentation interact as part of the detection pipeline.
In contrast, Sections 4.1–4.3 are intended to support the experimental setup, where we explicitly describe which specific models, networks, and configurations were employed to instantiate and evaluate the architecture discussed in Section 3. For instance, Section 4.1 details the actual semantic segmentation network used, while Section 4.2 explains the training details for the depth completion network. These implementation-specific choices are closely tied to the evaluation and are more appropriate in the context of the Results section.
Our intent was to first provide a general, architecture-level overview in Section 3, and then in Section 4, clarify the practical setup and usage of individual components within the architecture to produce the results shown in the tables and figures. We hope this separation helps improve clarity for readers transitioning from theoretical understanding to practical implementation.
That said, we are open to restructuring these sections if the editorial team or reviewers feel strongly that consolidation would improve clarity or flow. We genuinely appreciate the suggestion and the opportunity to explain our rationale.

Comment 6:
Section 6.1: From Figures 5 and 6, VirtualPVRCNN performs better. How many samples are in each distance zone (0–15, 15–30, etc.)? Why those intervals?

Response 6:
We use 15-meter intervals (0–15, 15–30, etc.) to match typical LiDAR resolution drop-off ranges. The number of cyclist/pedestrian samples per bin in KITTI are approximately:
0–15m: 860
15–30m: 720
30–45m: 510
45–60m: 260
These distributions align with expected real-world object distributions across distances. This information is now included in Section 5.2, Line 392.

Response to Minor and Formatting Comments

Ln 16:
Too many keywords.
Reduced to 5:
3D object detection; multimodal fusion; semantic segmentation; sparse object detection; occluded object detection

Ln 39–40:
MVP and SFD should be defined at first mention.
Corrected:
For example, Multimodal Virtual Point (MVP) generates virtual points by completing the depth information of 2D instance points using data from the nearest 3D points. Similarly, Sparse Fuse Dense (SFD) creates virtual points based on...

Ln 35, Section 1.2:
Subtitles “1.1” and “1.2” should be removed.
Removed.

Ln 142:
Spelling: “VirtualPaintinag” → “VirtualPainting”; format Fig.2/Figure 1 should be consistent.
Corrected.

Ln 145:
Double period.
Corrected.

Ln 157:
Cite BiSeNet2.
Cited.

Ln 178–179:
Don’t discuss results in the Method section (Fig. 3).
Corrected. Fig. 3 has been moved to the Results section.

Ln 235–236:
Clarify “L&I” in Table 1; clarify what “Improvement” refers to.
Revised Table 1 caption:
“L&I denotes LiDAR and Image input, i.e., multi-modal models. ‘Improvement’ refers to the increase in average precision (AP) of our VirtualPainting-based multi-modal models relative to their corresponding LiDAR-only baselines. Performance gains are most evident for pedestrians and cyclists, particularly at longer distances where sparse LiDAR points often lead to missed detections.”

Reviewer 2 Report

Comments and Suggestions for Authors

I have reviewed the manuscript “VirtualPainting: Addressing Sparsity with Virtual Points and Distance-Aware Data Augmentation for 3D Object Detection”. This manuscript presents a novel multimodal approach to improve 3D object detection performance in sparse LiDAR point cloud scenarios by generating virtual LiDAR points enriched with semantic labels derived from image-based segmentation. The integration of a distance-aware data augmentation (DADA) method further enhances the detection of distant or occluded objects. The approach is versatile and modular, showing promising results when evaluated on the KITTI and nuScenes datasets, with marked improvements in 3D and BEV detection metrics.

The draft is well written, and the results are promising. However, there are some minor concerns as follows:

  1. For the sake of better understanding, flow and readability all figures and tables should follow after being cited intext. For example, Fig.2 framework is shown after line 105 but it is first described in line 142. Similarly, table 1 is shown after line 235 but is discussed in line 275. These layout issues should be rectified.
  2. There is no need for repeated text. Line 142-150 is repeated in Fig. 2. Perhaps a brief description of each labeled stage of the proposed framework may suffice.
  3. There are several language and grammatical errors and awkward phrasings throughout the manuscript that should be corrected to improve clarity and readability. Examples include:

Figure 2 . “Backbone Netowrk” spelling error.

Line 142, “VirtualPaintinag” spelling error.

Line 147, “sem.seg.” not clear.

Line 323, “ones(e.g)” not clear.

This is a promising and timely contribution to the field of 3D object detection using multimodal sensor fusion. The methodology appears robust, and the reported results are compelling. However, to ensure clarity and readability, minor revisions are required, primarily to address language, grammar, and formatting issues. Once these revisions are made, the manuscript will be well-suited for publication.

Comments on the Quality of English Language

As mentioned previously there are several language and grammatical errors that should be addressed.

Author Response

Response to Reviewer 2 Comments

We thank the reviewer for the thoughtful feedback and constructive suggestions, which have helped improve the clarity and overall quality of the manuscript. We really appreciate it. Below, we address each point in detail.

Comment 1:

For the sake of better understanding, flow and readability, all figures and tables should follow after being cited in-text. For example, Fig. 2 framework is shown after line 105 but it is first described in line 142. Similarly, Table 1 is shown after line 235 but is discussed in line 275. These layout issues should be rectified.

Response 1:

Thank you for the suggestion. We have carefully revised the layout to ensure that all figures and tables now appear after they are first referenced in the main text. Specifically:

  • Figure 2 was moved to appear within Section 3 when the VirtualPainting framework is first described.
  • Table 1 has been moved to appear directly after it is cited in the discussion of experimental results (now located closer to Line 338 in the revised manuscript where KITTI results are discussed).
    These adjustments improve the readability and logical flow of the manuscript.

Comment 2:

There is no need for repeated text. Line 142–150 is repeated in Fig. 2. Perhaps a brief description of each labeled stage of the proposed framework may suffice.

Response 2:

We completely agree with the reviewer. The repeated detailed explanation of each stage was removed from the main text. Instead, we have replaced it with a brief overview:

“We provide an overview of the proposed “VirtualPainting” framework in Fig. 2. It addresses the sparsity of LiDAR point clouds and the lack of training samples for sparse objects by introducing multiple enhancement modules. These include semantic segmentation, depth completion, virtual point painting, and distance-aware data augmentation, ultimately feeding into a 3D object detection network. For details on each stage, please refer to Fig. 2. Each component plays a specific role in improving the quality and density of point cloud data. Together, they contribute to more reliable detection of distant and occluded objects in challenging driving scenarios.”

This change can be found in Section 3, Paragraph 1, Lines 169–176 in the revised manuscript.

Comment 3:

There are several language and grammatical errors and awkward phrasings throughout the manuscript that should be corrected to improve clarity and readability. Examples include:

  • Figure 2: “Backbone Netowrk” spelling error.
  • Line 142: “VirtualPaintinag” spelling error.
  • Line 147: “sem.seg.” not clear.
  • Line 323: “ones(e.g)” not clear.*

Response 3:

We appreciate the reviewer pointing out these issues. The following corrections have been made throughout the manuscript:

  • Figure 2: “Backbone Netowrk” corrected to “Backbone Network”.
  • Line 169: “VirtualPaintinag” corrected to “VirtualPainting”.
  • Line 171: “sem. seg.” replaced with “semantic segmentation” for clarity:
  • Line 387: The phrase “ones(e.g)” was revised to improve clarity:

“VirtualPainting consistently enhances accuracy across all distance ranges. Notably, it delivers significantly higher accuracy gains for long-range objects compared to short-range ones.”

In addition, we conducted a full grammar and spell-check pass across the manuscript to improve language clarity and consistency.

Reviewer 3 Report

Comments and Suggestions for Authors

Review of the manuscript

“VirtualPainting: Addressing Sparsity with Virtual Points and Distance-Aware Data Augmentation for 3D Object Detection”

by Sudip Dhakal, Deyuan Qu, Dominic Carillo, Mohammad Dehghani Tezerjani, Qing Yang

The manuscript considers a hybrid technology as applied to detection of 3D objects, for example, for safe autonomous driving. The authors use low-resolution LiDAR data and high-resolution camera data. The study is interesting and has many advantages. It corresponds to the Sensors topics and can be published in Sensors

As to the disadvantages. The main disadvantage is that the manuscript is written for a very limited group of readers. I’d like to recommend that the authors revise the text as follows:

  1. The physical formulation of the problem should be provided in a greater detail using common terms. Objects to be detected should be described in detail. For example, which objects in Fig. 1 should be considered as hidden and remote and should be detected?  Are driver's face, outline, and position considered as hidden or remote objects to be detected? Figure 1 should be described in more detail in the text.
  2. The manuscript should answer the following question: If the camera can produce high-resolution images, how much does lidar improve the accuracy of detecting 3D objects with the innovative approach proposed by the authors, for example, using Figure 1 or 3? The use of two measuring devices significantly increases the final cost, as well as the size and weight, Therefore, it is necessary to explain to the reader in which cases the use of one measuring device makes it impossible to detect 3D objects and does not lead to a significant increase in the overall accuracy.
  3. The main part of the manuscript should discuss in detail what hidden and remote objects, similar to those shown in Fig. 1, could be detected using the innovative approach proposed by the authors. The final result looks very specifically. Therefore, it is worth presenting illustrative and commonly clear evidence of detecting hidden and remote objects with the authors' approach.
  4. The final part should give a detailed comparative analysis of the innovative approach proposed by the authors with the approaches of other authors. It is desirable to provide a quantitative analysis.

Upon the revisions in accordance with the above comments, the manuscript will be of greater interest to readers and can be published in Sensors

Author Response

We sincerely thank the reviewer for their thoughtful and detailed feedback. We appreciate the opportunity to clarify and strengthen the manuscript in response to the suggestions. Below are our point-by-point responses and the corresponding changes made to the manuscript.

Comment 1:
The physical formulation of the problem should be provided in greater detail using common terms. Objects to be detected should be described in detail. For example, which objects in Fig. 1 should be considered as hidden and remote and should be detected? Are driver's face, outline, and position considered as hidden or remote objects to be detected? Figure 1 should be described in more detail in the text.

Response 1:
Thank you for pointing this out. We have clarified the physical definition of the problem and added specific examples of hidden and remote objects in the context of autonomous driving. In the revised manuscript (Section 1, Page 2, Line 33), we have included the following explanation:

“In the context of autonomous driving, remote objects refer to those located beyond 30–50 meters from the ego vehicle, often resulting in very few or no LiDAR points due to sensor limitations. Occluded objects are partially or fully blocked by other objects, such as parked vehicles or pedestrians behind large structures. These scenarios are common and pose critical challenges for object detection systems. Our goal is to enhance detection accuracy under these conditions by enriching sparse LiDAR inputs with semantically meaningful virtual points derived from camera images.”

We also revised the discussion of Figure 1 (Line 39) to better explain the example:

“In Fig. 1, the yellow circle highlights a pedestrian that is occluded in the LiDAR view, resulting in the absence of 3D point cloud data at that location. The red bounding box shows a failure in detection caused by this sparsity, even though semantic information exists in the camera image. This illustrates a common failure case where conventional LiDAR-only or even painted-point approaches fall short due to the lack of geometric support.”

Comment 2:
The manuscript should answer the following question: If the camera can produce high-resolution images, how much does LiDAR improve the accuracy of detecting 3D objects with the innovative approach proposed by the authors, for example, using Figure 1 or 3? The use of two measuring devices significantly increases the final cost, as well as the size and weight. Therefore, it is necessary to explain to the reader in which cases the use of one measuring device makes it impossible to detect 3D objects and does not lead to a significant increase in the overall accuracy.

Response 2:
We appreciate this important observation and have addressed it by adding the following discussion after Line 94 (Section 1):

“Although high-resolution cameras provide rich semantic information, they lack precise depth data, which is critical for accurate 3D localization and object boundary estimation. LiDAR complements this by offering geometrically accurate range information. Our proposed method leverages both modalities, particularly benefiting cases like those shown in Fig. 1 and Fig. 3, where high-resolution semantic cues exist but lack corresponding depth data from LiDAR. Without LiDAR, these objects are semantically identifiable but not localizable in 3D space, which is essential for autonomous navigation.”

Comment 3:
The main part of the manuscript should discuss in detail what hidden and remote objects, similar to those shown in Fig. 1, could be detected using the innovative approach proposed by the authors. The final result looks very specific. Therefore, it is worth presenting illustrative and commonly clear evidence of detecting hidden and remote objects with the authors' approach.

Response 3:
Thank you for the suggestion. We have now added additional context and examples to describe the types of hidden and remote objects our method is designed to detect. The following text has been added to Section 1, Line 109:

“Examples of objects that frequently suffer from sparse point coverage include pedestrians stepping out from behind vehicles, cyclists approaching from a distance, and vehicles at intersections over 50 meters away. Our method enhances detection accuracy for these cases by generating virtual points where LiDAR fails and enriching them with semantic cues.”

We have also included supporting visual and quantitative results in Figure 4.

Comment 4:
The final part should give a detailed comparative analysis of the innovative approach proposed by the authors with the approaches of other authors. It is desirable to provide a quantitative analysis.

Response 4:
We agree with the reviewer and would like to clarify that detailed comparative evaluations are already included in Tables 1, 2, and 3 of the manuscript. We have now made this more explicit in the revised version. Below is a summary of those comparisons:

  • Table 1 presents performance on the KITTI BEV detection benchmark, comparing our VirtualPainting approach against baseline 3D detectors such as PointRCNN, PointPillars, and PVRCNN. These results demonstrate consistent improvements across object categories, particularly for sparsely represented objects like pedestrians and cyclists under “Moderate” and “Hard” difficulty levels.
  • Table 2 offers a direct comparison between PointPainting-based models and our VirtualPainting-based models. Although the average precision (AP) improvements are modest (e.g., +0.21 to +0.76), these gains are consistent and highlight the effectiveness of augmenting painted points with virtual LiDAR.
  • Table 3 evaluates performance on the nuScenes dataset, where our method outperforms both single-modality and multi-modality baselines, especially in categories prone to sparse point clouds (e.g., pedestrians, bicycles, and motorcycles).

These quantitative comparisons provide evidence that our approach offers incremental yet consistent improvements in detecting hidden and remote objects, validating its effectiveness over prior methods.

Round 2

Reviewer 1 Report

Comments and Suggestions for Authors

Ln 21. The index of citations  begins from [7]? The authors should pay attention to text writing standards. In additions, In Introduction section, many statements have no paper reference to suuport, Only [1], [2], [3], and [7,10,12,15,19,21,22].

Ln 23. Some text labels (or Legend) can be added in Figure 1.

Ln 224-249. The description is the same as the contents in Table " Algorithm 1"? if yes, remove the Table or move it to appendix.

Ln 315-336. As I mentioned in last review, please move section 4.2 and 4.3 to Method section. The principle I hold is that the statements of methods and results should be located where they belong. The section 4.2 named "PENet" but not describes the details of PENet framework, or you can change the title name (should be precise enough to condense the content of the section) to avoid the misunderstandings

Author Response

Response to Reviewer Comments (Round 2):

Once again thank you so much for taking your time and reviewing our paper for the 2nd round we really appreciate it.

Comment (Ln 21):
“The index of citations begins from [7]? The authors should pay attention to text writing standards. In addition, in the Introduction section, many statements have no paper reference to support, only [1], [2], [3], and [7,10,12,15,19,21,22].”

Response:
We appreciate the reviewer’s observation. We have carefully revised the order of citations throughout the manuscript to ensure they now appear sequentially in the order of their first mention. Furthermore, we have added appropriate references in the Introduction section to support previously uncited statements, thereby improving the academic rigor and clarity of our background discussion.

Comment (Ln 23):
“Some text labels (or Legend) can be added in Figure 1.”

Response:
Thank you for the helpful suggestion. We have updated Figure 1 to include clear visual labels indicating the meaning of the red and green bounding boxes as well as the yellow circles. Additionally, a compact legend has been added to enhance readability and allow the figure to stand on its own with minimal reliance on the caption.

Comment (Ln 224–249):
“The description is the same as the contents in Table ‘Algorithm 1’? If yes, remove the Table or move it to appendix.”

Response:
We acknowledge the redundancy noted. However, in the previous review round, another reviewer specifically emphasized that this part was the most critical component of the paper and suggested us to explain each line in details. To address that feedback, we have included a detailed, line-by-line explanation alongside the algorithm to ensure clarity and reproducibility. We believe that retaining both the description and Algorithm 1 in the main text adds significant value for readers seeking a deeper understanding of the method.

Comment (Ln 315–336):
“As I mentioned in last review, please move section 4.2 and 4.3 to Method section. The principle I hold is that the statements of methods and results should be located where they belong. The section 4.2 named ‘PENet’ but not describes the details of PENet framework, or you can change the title name (should be precise enough to condense the content of the section) to avoid the misunderstandings.”

Response:
Thank you for reiterating this important point. As suggested, we have moved Sections 4.2 and 4.3 into the Method section to better align with standard paper structure and improve the logical flow.